# Differential Production of Nitric Oxide and Hydrogen Peroxide among *Drosophila melanogaster*, *Apis mellifera*, and *Mamestra brassicae* Immune-Activated Hemocytes after Exposure to Imidacloprid and Amitraz

**DOI:** 10.3390/insects14020174

**Published:** 2023-02-09

**Authors:** Dani Sukkar, Philippe Laval-Gilly, Antoine Bonnefoy, Sandhya Malladi, Sabine Azoury, Ali Kanso, Jairo Falla-Angel

**Affiliations:** 1Biology Department, Faculty of Sciences I, Lebanese University, Hadath 1003, Lebanon; 2Laboratoire Sols et Environnement, Institut National de Recherche pour l’Agriculture, l’Alimentation et l’Environnement (INRAE), Université de Lorraine, 54000 Nancy, France; 3Plateforme de Recherche, Transfert de Technologie et Innovation (PRTI), Institut Universitaire de Technologie de Thionville-Yutz, Université de Lorraine, 57970 Yutz, France

**Keywords:** invertebrate immunity, humoral response, inflammatory response, insects, pesticides, neonicotinoids

## Abstract

**Simple Summary:**

Pesticide exposure is a risk factor affecting insect immune response and may participate in honeybee colony deaths. Changes in the immune response may hinder insects more susceptible to diseases and increase the risk of pathogen infection and pest resistance. We found that imidacloprid and amitraz alter the oxidative response of insect hemocytes at different concentrations of the immune stimulator, zymosan A. There is a differential effect of imidacloprid and amitraz on hemocytes of insects from different orders and their effect on oxidative response in insects persists beyond the point of early contact with zymosan A.

**Abstract:**

Invertebrates have a diverse immune system that responds differently to stressors such as pesticides and pathogens, which leads to different degrees of susceptibility. Honeybees are facing a phenomenon called colony collapse disorder which is attributed to several factors including pesticides and pathogens. We applied an in vitro approach to assess the response of immune-activated hemocytes from *Apis mellifera*, *Drosophila melanogaster* and *Mamestra brassicae* after exposure to imidacloprid and amitraz. Hemocytes were exposed to the pesticides in single and co-exposures using zymosan A for immune activation. We measured the effect of these exposures on cell viability, nitric oxide (NO) production from 15 to 120 min and on extracellular hydrogen peroxide (H_2_O_2_) production after 3 h to assess potential alterations in the oxidative response. Our results indicate that NO and H_2_O_2_ production is more altered in honeybee hemocytes compared to *D. melanogaster* and *M. brassicae* cell lines. There is also a differential production at different time points after pesticide exposure between these insect species as contrasting effects were evident with the oxidative responses in hemocytes. The results imply that imidacloprid and amitraz act differently on the immune response among insect orders and may render honeybee colonies more susceptible to infection and pests.

## 1. Introduction

The effect of pesticide application on the immune response in insects has been a subject of importance in invertebrate immunity studies and environmental risk assessments. Co-exposure to pesticides and diseases dysregulates the immune response and produces a synergistic effect on pollinator insect decline [1,2,3,4]. Anthropogenic activity and exploitation of the Earth’s resources are designated as the causes of species extinction equivalent to the past five extinction events; in particular, the “insectageddon”, as reviewed by [5]. The concept is supported by the fact that the rate of insect deaths is 8 times faster than mammals, birds, and reptiles [6]. Pollinator insects, specifically, have a global impact since they are responsible for pollinating over 80% of the world’s crops [7]. Pollinator-mediated pollination contributes to the maintenance of the biodiversity of plants that depend on insects for reproduction and species continuation [8,9]. An important single pollinator species with high economic value is the European honeybee (*Apis mellifera*) [7,10]. Honeybees are exposed to multiple environmental factors and stressors that are suspected to contribute to a phenomenon known as colony collapse disorder (CCD) characterized by the sudden disappearance of honeybee workers leaving the brood and the queen unattended, eventually leading to the death of the colony [11]. In a meta-genomic study conducted by Cox-Foster et al. [12], CCD was speculated to be a result of honeybee exposure to several interactive factors leading to colony loss.

Factors that are implicated in the pollinator population decline include pests, diseases, and pesticides. Of the studied diseases, the microsporidian fungus *Nosema cerenae* was found in 100% of CCD colonies in the USA and Australia [12]. *Nosema* spp. are the causal agent of nosemosis diseases and mainly infect the bee ventriculus, preventing nutrient uptake and ultimately leading to the death of the host [13,14,15]. The biotic challenges to honeybees do not cease with nosemosis, as weaker colonies are better targets for the pest known as the wax moth. The lesser wax moth (*Achroia grisella*) and the greater wax moth *(Galleria mellonella)* are both pests that infiltrate beehives where their larvae parasitise on the brood, thereby damaging the colony [16,17].

As for the pesticides of most concern, attention is drawn to neonicotinoids which dominated the global market in 2010 [18]. Imidacloprid, especially, is stated to be the most used neonicotinoid in the world [19,20]. Neonicotinoids are systemic neuroactive insecticides, acting on the central immune system and irreversibly blocking nicotinic acetylcholine receptors (nAChRs) [21]. Though the main targets of neonicotinoids are insect pests such as sucking insects, aphids, hoppers, and thrips, non-target species are also affected [22].

Other than neonicotinoids, amitraz is an acaricide and insecticide that is in direct contact with honeybees. It is an octopamine receptor agonist used to control the honeybee ectoparasitic mite, *Varroa destructor.* The presence of amitraz was reported to produce a synergistic effect when present with other pesticides by inhibiting their P450-mediated detoxification while the toxicity of amitraz itself remained unchanged [23].

Protection against pathogens is facilitated by the production of reactive nitrogen species (RNS) and reactive oxygen species (ROS) via humoral-mediated immune response [24] and/or cellular processes including autophagy and immune signaling and regulation of immune processes [25]. Nitric oxide (NO) was found to be produced in the primary immune response and acts as a signaling molecule to downstream immune pathways and is also affected by pesticide exposure [26,27,28]. The second function of nitric oxide is a defense against pathogens [29]. Hydrogen peroxide is an ROS that is produced as a defensive response of the innate immune system [30] as the first line of defense with RNS [31]. Nitric oxide and hydrogen peroxide were found to participate in the induction of systemic immune response in insects such as mosquitos [32]. Neonicotinoids were found to increase hydrogen peroxide production, leading to oxidative stress and DNA damage in the Dipteran aquatic insect *Chironomus dilutus* through mitochondrial Ca^+^ flux [33]. Interestingly, imidacloprid was found to reduce hydrogen peroxide production in honeybees, bumblebees [34], and *Drosophila* [35]. Variation in the immune response to pesticide exposure between and within different orders necessitates the assessment of species-specific immune responses. The variability of the diverse immune system in insects [36] extrapolates to variable immune responses concerning the pesticide–pathogen interaction. It is already evident that the effect of pesticide exposure on the immune response in insects is variable between different orders [37] and within the same order as in the case of honeybees and bumblebees [34].

In this study, we took an in vitro approach via cell-based assays to assess the effect of imidacloprid and amitraz on insect immune cells “hemocytes” and their production of nitric oxide and hydrogen peroxide. Along with *A. mellifera* hemocytes, we included the Schneider-2 cell line established from *Drosophila melanogaster* (Diptera) hemocytes. *Drosophila* was previously used as a model organism to study interactions with neonicotinoids and pathogens [35,38,39,40]. We also included the MB-L2 cell line established from larvae of the cabbage moth (*Mamestra brassicae*) as a model for hemocytes of Lepidopteran insects such as the wax moth. Thus, we will take a comparative approach between these 3 insect orders to (1) explore how pesticides affect the immune response hemocytes of Lepidoptera compared to honeybees, and (2) compare the immune response of the model organism, *D. melanogaster’s* Schneider-2 organism cell line to Lepidopteran cell line, and to freshly extracted hemocytes of honeybees.

The assessment of the presence of multiple pesticides and pathogens simultaneously is a necessity to understand the interplay of these factors in CDD. Combined exposures were used in different ratios to evaluate the effect of the co-exposure of imidacloprid and amitraz in variating proportions and the effect of their possible interaction in an in vitro system. The cells were stimulated by zymosan A, a beta-glucan found as a cell wall component of *Saccharomyces cerevisiae* [41]. Zymosan A exposure was used for mimicking the immune activation induced by *Nosema* spp. infection. The beta-glucan was shown to elicit an immune response in insects and induce melanization [42], alter the immune–gene expression in the Toll pathway [43] and increase in the production of antimicrobial peptides [44].

In vitro systems provide insight into the effects of pesticides and immune responses in specific cell targets in controlled conditions with limited environmental variabilities that are not observable in in vivo systems. Although the in vitro system, individually, gives a partial view that cannot be completely extrapolated to the whole organism level, yet, it paves the way to a more comprehensive view on the effects observed in vivo. This work is part of an international effort to develop a better understanding of the potential causes of CCD. It also comprises a comparative immunological approach that may shed light on the necessity of exploring species-specific interactions and immune responses on the cellular level.

## 2. Materials and Methods

### 2.1. Cell Lines

The *D. melanogaster* Schneider-2 cell line (ACC 130) and the cabbage moth (*M. brassicae*) MB-L2 cell line (ACC 76) were purchased from Leibnitz-Institute DSMZ Deutsche Sammlung von Mikroorganismen und Zellkulturen GmbH (Braunschweig, Germany). Honeybee (*A. mellifera*) hemocytes were freshly extracted from larvae before every experiment as described below.

### 2.2. Hemocyte Extraction

Fifth instar larvae (*A. mellifera*) were collected from hives established at IUT Thionville/Yutz, France. Under a laminar flow hood, each larva was gently held between the index finger and thumb of the non-dominant hand exposing the dorsal posterior part, which in turn was sterilized with a cotton swab dipped in 70% ethanol. The sterilized area was punctured by a sterile needle, hemolymph droplet was quickly collected by a micropipette that was set at 35 µL and then transferred to a tube containing WH2 medium or PBS depending on the experiment.

### 2.3. Culture Medium and Maintenance

Hemocytes were maintained in their respective culture mediums at 20 °C with no carbon dioxide exposure in 25 cm^2^ tissue culture flasks (90025, TPP^®^, Trasadingen, Switzerland). Schneider-2 hemocytes were maintained in a culture medium containing 44.5% Schneider’s Drosophila medium (21720-024, Gibco™, Billings, MT, USA), 44.5% TC-100 insect medium (T3160, Sigma Aldrich™, Burlington, MA, USA), 10% de-complemented fetal bovine serum (F7524, Sigma Aldrich™, Gillingham, UK), and 1% penicillin-streptomycin (P4458, Sigma Aldrich™, Burlington, MA, USA). Grace’s insect medium (G8142, Sigma Aldrich™, Burlington, MA, USA), supplemented with 10% de-complemented fetal bovine serum and 1% penicillin-streptomycin, was used to maintain MB-L2 cells. For honeybee hemocytes, we used WH2 medium, as described by [45]. The WH2 medium was composed of 35.55% Schneider’s insect medium, 47.39% 0.06 M L-Histidine (H5659, Sigma Aldrich™, St. Louis, MO, USA), 11.85% de-complemented fetal bovine serum, 3.55% CMRL 1066 (11530037, Gibco™, Paisley, UK), 1.18% Hank’s salt solution (H6648, Sigma Aldrich™, Burlington, MA, USA), 0.47% insect medium supplement 10× (I7267, Sigma Aldrich™, St. Louis, MO, USA), and 1% gentamicin (G1397, Sigma Aldrich™, Burlington, MA, USA). The pH of the WH2 medium was regulated to 6.35–6.5 using the 2N NaOH solution then sterile filtered with a 0.2µm low-protein-binding syringe filter (Acrodisk™ 4312, Pall Corp. ™, Port Washington, NY, USA).

### 2.4. Exposures

We applied a maximum concentration of 50 µg/mL for both active ingredients, amitraz and imidacloprid, according to similar concentration limits used in the literature for laboratory experiments including cell-based and organism systems [34,46,47]. Hemocytes are not the main targets of imidacloprid or amitraz and thus we used concentrations that exceed the realistic field levels in order to obtain a detectable effect. Imidacloprid (37894-100 mg, Sigma-Aldrich™, Basel, Switzerland) and zymosan A (Z4250, Sigma-Aldrich™, Basel, Switzerland) stock solutions of concentrations of 1 mg/mL were made by dissolving the powdery compound in Dulbecco’s Phosphate Buffered Saline (PBS; D8537, Sigma-Aldrich™ St. Louis, MO, USA) via sonication. Amitraz (45323, Sigma-Aldrich™, Basel, Switzerland) was dissolved in hexane at 10 mg/mL. All chemicals were diluted in a culture medium or PBS depending on the experiment and amitraz was left for 1 h to allow for hexane to evaporate before cell exposure. The final concentrations of imidacloprid and amitraz were 10 and 50 µg/mL. Mixture solutions included 10 + 10 µg/mL, 10 + 50 µg/mL or 50 + 10 µg/mL of imidacloprid and amitraz, respectively. Hemocytes in all treatments were either immune-activated by 1 or 10 µg/mL of zymosan A or with no immune activation.

### 2.5. Viability Assay

A viability assay was performed to assess the applicability of the assays with the used concentrations. Cultured cell lines were detached by gentle pipetting to not affect the viability outcome. Hemocytes were seeded in 96-well plates containing the treatments in a culture medium. The plates were sealed with sealing tape and incubated with the treatments for 18 h at 20 °C. The duration was chosen to allow for honeybee hemocytes to stabilize in the culture medium after extraction so as not to skew the results of the assay. Viability was assessed using trypan blue dye exclusion [48]. Trypan blue (T4146, Sigma-Aldrich™, Basel, Switzerland) was dissolved in 0.9% NaOH solution (S/4800/65, Fischer Scientific™, Loughborough, UK) at a 2 g/mL concentration. In each well, we added trypan blue in a 1:1 volumetric ratio. After 3 min of incubation, at least 200 cells were counted under an inverted microscope in each well. Each condition was carried out in triplicates. Hemocytes that retained the dye within them were considered dead. Cell viability percentage was calculated using the following equation for each well:$$Percentage\;of\;viable\;cells\;\%=\frac{total\;number\;ofcounted\;cells-total\;number\;of\;dead\;cells\;}{total\;number\;ofcounted\;cells}\times 100$$

### 2.6. Nitric Oxide Quantification

Treatments were added to a 96-well Half Area High Content Imaging Glass Bottom Microplate (4850, Corning^®^, Corning, NY, USA). Nitric oxide (NO) production was measured using DAF-FM DA fluorescent dye (D1946, Sigma-Aldrich™, Tokyo, Japan), as carried out by [28]. A stock solution of 50 mM was dissolved in DMSO. Exposure mixtures that included pesticides and zymosan were prepared in PBS and added to the plates before seeding. Hemocytes were centrifuged at 1000 rcf for 5 min and washed with PBS twice before the addition of the DAF-FM DA to the hemocytes with a concentration of 20 µM just before plate seeding. The final concentration of DAF-FM DA was 5 µM in each well while the final concentrations of pesticides were 10 and 50 µg/mL for imidacloprid and amitraz single exposures and 10 + 10 µg/mL, 10 + 50 µg/mL or 50 + 10 µg/mL of imidacloprid and amitraz, respectively, for mixtures. All pesticide exposures were either carried out with 0, 1 or 10 µg/mL zymosan A. All treatments included 5 or 6 replicates. Nitric oxide production was measured at 15 min of exposure until 120 min by fluorescence detection via Fluostar Galaxy (BMG Labtech™, Sorisole, Italy) fluorimeter with an excitation/emission of λ = 490/515 nm. The following equations were used to represent the obtained data:$$Nitric\;oxide\;production\;at\;15\;min\;\left(\%\right)=\frac{fluorescence\;of\;sample\;at\;15\;min}{\begin{array}{c} Avg.fluorescence\;of\;the\;control\;group\;\\ without\;zymosan\;at\;15\;min\; \end{array}}\times 100$$
$$Nitric\;oxide\;production\;at\;120\;min\;\left(\%\right)=\frac{fluorescence\;of\;sample\;at\;120\;min}{\begin{array}{c} Avg.fluorescence\;of\;the\;control\;group\\ \;without\;zymosan\;at\;120\;min\; \end{array}}\times 100$$

The rate of nitric oxide production from 15 min to 120 min was calculated by using the equations:(1)$$Sample\;ratio=\frac{fluorescence\;of\;sample\;at\;120\;min}{fluorescence\;of\;sample\;at\;15\;min}$$
(2)$$Control\;group\;ratio=\frac{\begin{array}{c} Avg.fluorescence\;of\;the\;control\;group\\ \;without\;zymosan\;at\;120\;min\; \end{array}}{\begin{array}{c} Avg.fluorescence\;of\;the\;control\;group\\ \;without\;zymosan\;at\;15\;min\; \end{array}}$$
$$Change\;in\;nitric\;oxide\;production\;rate\;\left(\%\right)=100\times \left[\frac{\mathrm{Equation}\left(1\right)}{\mathrm{Equation}\left(2\right)}-1\right]$$

### 2.7. Hydrogen Peroxide Assay

Hemocytes were plated in 96-well plates containing treatments in their respective culture mediums in a total volume of 70–100 µL with 5 replicates of each treatment. The plates were sealed with sealing tape and incubated for 3 h at 20 °C. Extracellular hydrogen peroxide was quantified using Amplex™ Red Hydrogen Peroxide/Peroxidase Assay Kit (Invitrogen™, Waltham, MA, USA, catalog no. A22188). A 100 μM Amplex^®^ Red reagent and 0.2 U/mL HRP working solution in 1× reaction buffer were prepared according to the manufacturer’s protocol. Hydrogen peroxide standards were prepared in 1× reaction buffer with concentrations between 0 and 10 µM. Plates were centrifuged using ThermoScienfic™ Megafuge 16R centrifuge at 1000× *g* rcf for 3 min, and 50 µL of the supernatant was transferred to a new 96-well plate. The working solution was transferred in 50 µL volume to each well containing the supernatant and the hydrogen peroxide standards, then incubated for 30 min in the dark. Absorbance was measured using a spectrophotometer (BioTek™, Winooski, VT, USA, PowerWave XS2) at λ = 560 nm and the results were normalized to the absorbance of the culture medium, expressed as relative to the control group with no pesticide exposure.

### 2.8. Statistical Analysis

Statistical analysis was carried out using XlSTAT^©^ 2019 v21.3.62913.0 (Addinsoft™, Long Island, NY, USA). Normality was tested by the Shapiro–Wilk test for each group of zymosan A treatments. To compare pesticide treatments, groups of normal distribution were tested for significance using ANOVA (Dunnett’s test or Tukey HSD). Groups with no normal distribution were tested for significance using Kruskal–Wallis and Dunn’s pairwise comparison.

## 3. Results

### 3.1. Immune Activation Alleviates the Effect of Pesticides on Hemocyte Viability

The cell lines of Schneider-2 and MB-L2 showed no observable decrease in viability via the trypan blue exclusion method with 100% viability in all conditions. As for *A. mellifera* hemocytes, the results are shown in (Figure 1). Viability was 97.55% for the control group with no immune activation and similar viability of 97.60% and 97.68% with immune activation by 1 and 10 µg/mL of zymosan A (the statistical significance between groups is not shown). Within the group of no zymosan exposure, 10 µg/mL of imidacloprid or the pesticide combinations of either higher imidacloprid or higher amitraz combinations resulted in significantly higher viability of 99.25% compared to the no-pesticide control.

The exposure to amitraz with no immune activation resulted in a significant decrease in viability to 95.91% compared to all pesticide exposures except the control group. All conditions that included immune activation with zymosan A at 1 and 10 µg/mL showed no significant difference in viability.

### 3.2. Nitric Oxide Endpoint Production

Nitric oxide was measured by fluorescence intensity at 15 min (Figure 2) and 120 min (Figure 3). Honeybee (*A. mellifera*) hemocytes produced significantly less nitric oxide reaching 73.55, 73.13 and 74.17% after 15 min incubation with no immune activation (Figure 2A) when exposed to 10 and 50 µg/mL amitraz or the 50I-10A combination, respectively. A higher significant decrease to 69.85% and 67.06% was observed when exposed to 50 µg/mL imidacloprid or the 10I-10A combination. When honeybee hemocytes were immune-activated with 1 µg/mL zymosan A (Figure 2B) all pesticide exposures revealed a decrease in nitric oxide production after 15 min compared to the no-pesticide control of the group.

Nevertheless, significance was only apparent with 50 µg/mL imidacloprid, 10 and 50 µg/mL amitraz, and the 10I-10I combination at 79.52, 81.49, 78.53 and 80.2% NO production, respectively. Interestingly, high immune activation with 10 µg/mL zymosan A showed no significant difference with any of the treatments (Figure 2C). There was no significant effect on nitric oxide production with or without immune activation by zymosan A for the Schneider-2 (Figure 2D–F). However, the MB-L2 cell line showed a significant decrease to 92.27, 89.88 and 89.54% in nitric oxide production after 15 min of exposure to pesticide mixtures 10I-10A, 10I-50A, and 50I-10A, respectively, at high immune activation (Figure 2I), while no significant change was detected with no zymosan exposure or with 1 µg/mL zymosan A (Figure 2G,H). The effect of the treatments on nitric oxide production was more apparent after 120 min of exposure (Figure 3). The decrease in nitric oxide production in honeybee (*A. mellifera*) cells was highly significant in all the treatments compared to the control in the 0 zymosan A group (Figure 3A) with a 62.87% decrease when hemocytes were exposed to the 10I-50A combination. In the same group, the maximal decrease was to 46.99% when hemocytes were exposed to 50 µg/mL imidacloprid. This pattern is even more pronounced when honeybee cells were immune-activated with 1 µg/mL zymosan A (Figure 3B), with the least decrease to 70.56% with 10 µg/mL imidacloprid reaching 56.03%, 55.94% and 56.21% with 50 µg/mL imidacloprid, 50 µg/mL amitraz and 10I-10A combination, respectively. However, with 10 µg/mL zymosan A immune activation (Figure 3C), the highest significant decrease in NO production in *A. mellifera* hemocytes was observable with the pesticide combinations followed by 10A, 50A, and 10I exposures while 50I exposure showed no significant change at 120 min.

Contrary to *A. mellifera* hemocytes, Schneider-2 cells showed no significant change in NO production at 120 min with no immune activation except for the 16.78% increase with the 50I-10A pesticide combination (Figure 3D). When challenged with 1 µg/mL zymosan A, Schneider-2 cells produced significantly less NO with 10I-50A combination and more NO with the 50I-10A combination (Figure 3E). Interestingly, with 10 µg/mL zymosan A, an increase in NO production in all pesticide treatments compared to the control was observed (Figure 3F). The significance was highest with amitraz single exposures (*p* < 0.01), while exposures that included imidacloprid in 10I, 50I, and 10I-50A showed an increase in NO with less significance (*p* < 0.05). The combinations 10I-10A and 50I-10A were not significantly different when compared to the Schneider-2 control at 120 min of exposure.

The Lepidopteran MB-L2 cell line showed a significant decrease of 32.13, 36.9, and 35.7% in NO production at 120 min without immune activation when exposed to pesticide combinations 10I-10A (*p* < 0.05), 10I-50A, and 50A-10I (*p* < 0.01) (Figure 3G). When immune-activated by 1 µg/mL zymosan A (Figure 3H), MB-L2 cells showed a pattern of response to pesticide exposures similar to that without immune activation. NO production was lower in higher concentrations of amitraz and imidacloprid and even lower in pesticide combinations but was only significant with the 10I-50A with a decrease to 65.65% at 120 min. Immune activation with 10 µg/mL zymosan A resulted in no change in NO production by MB-L2 cells at 120 min of exposure (Figure 3I).

### 3.3. Nitric Oxide Production Rate

We calculated the change in NO production from 15 min to 120 min. Hemocytes from *A. mellifera* showed a significant decrease in NO production rate in all treatments without immune activation relative to the control condition except 50I-10A (−14.03%) (Figure 4A). The decrease is by 21.43, 30.84, 25.72, 31.51, 24.15, 18.47% for 10I, 50I, 10A, 50A, 10I-10A, and 10I-50A, respectively. In the 1 µg/mL zymosan A treatment group (Figure 4B), all pesticide treatments were highly significant (*p* < 0.0001) compared to the control, ranging from a 21.29% decrease with 50I-10A to 30.2% with 50 µg/mL imidacloprid (50I). With high immune activation (Figure 4C), a significant decrease was apparent with amitraz exposures and pesticide mixtures 10I-10A and 10I-50A.

Schneider-2 cells showed no change in NO production rate in all conditions regardless of zymosan concentration or pesticide treatment (Figure 4D–F). MB-L2 cells showed a significant decrease in NO production rate without immune activation (Figure 4G) when exposed to pesticide mixtures 10I-10A, 10I-50A, and 50I-10A. No significant change in NO production rate was visible when MB-L2 cells were challenged with 1 or 10 µg/mL zymosan A (Figure 4H,I) except for the 50I-10A mixture with 10 µg/mL zymosan A (Figure 4I), of which showed an 11.25% increase.

### 3.4. Extracellular Hydrogen Peroxide Production

The production of hydrogen peroxide in *A. mellifera* hemocytes significantly decreased compared to the control with 10I-50A (41.87%) and 50I-10A (49.73%) when no zymosan A was applied (Figure 5A). When exposed to 1 µg/mL zymosan A (Figure 5B), all treatments showed a significant decrease compared to the control with the exception of 10A pesticide treatment. As for 10 µg/mL zymosan A group (Figure 5C), there was no statistically significant change in hydrogen peroxide production after 3 h. However, a concentration-dependent change was visible graphically.

A significant decrease to 55.58% in hydrogen peroxide production by Schneider-2 cells was observed in 50I-10A exposure with 1 µg/mL zymosan A (Figure 5E), and in all pesticide mixtures 10I-10A (62.14%), 10I-50A (66.15%) and 50I-10A (63.47%), but not in single exposures when exposed to 10 µg/mL zymosan A (Figure 5F) compared to the control within the group. Contrary to *A. mellifera* and Schneider-2 cells, the results from MB-L2 cells showed a significant increase in hydrogen peroxide production with no immune stimulation (Figure 5G). The higher concentration of amitraz (50A), mixtures 10I-50A and 50I-10A, have significantly higher H_2_O_2_ productions by 27.7, 38.64 and 27.7%, respectively. No significant change was observed in H_2_O_2_ production by MB-L2 cells with immune activation at either 1 and 10 µg/mL zymosan A (Figure 5H,I).

## 4. Discussion

No significant effect of pesticides on viability was detected on Schneider-2 or MB-L2 cell lines in all treatments, with no dead cells detected with trypan blue (100% viability). However, honeybee hemocytes surprisingly displayed significantly increased viability with the lower imidacloprid concentration or with a mixture of high and low concentrations of both pesticides compared to the no-pesticide control in the group without zymosan A exposure. The higher concentration of amitraz resulted in the lowest viability in non-immune-activated honeybee cells. This could be due to the increased production of enzymes that hydrolyze amitraz to the more potent metabolite DMFP [49]. Although the major target of amitraz, the Octβ2R octopamine receptor, is less sensitive to amitraz in honeybees than in target species, the metabolite DMFP can still bind to the receptor [50]. However, there was an antagonistic effect with treatments that included mixtures. A possible explanation would be that imidacloprid hindered either the production of enzymes associated with amitraz hydrolysis or their activity, leading to a reduced effect of amitraz exposure on viability. The same effect was absent in Schneider-2 and MB-L2 cells, possibly referring to a different impact on pesticide metabolism. Imidacloprid toxicity was found to be dependent on the production of the detoxifying enzymes of the cytochrome P450 rather than just the diversity of detoxifying enzymes [19]. Adding that amitraz toxicity is affected by its metabolism by cytochrome P450 [51] implies that the effect of imidacloprid and amitraz co-exposure may be intrinsically different from the effect of single exposures.

Zymosan A appears to mitigate the effect of imidacloprid and amitraz on primary NO production at 15 min with complete masking of the effect at the higher concentration of 10 µg/mL. The longer exposure of imidacloprid and amitraz affected NO production even further. At 120 min, all pesticide treatments decreased NO production regardless of the level of immune activation with the exception of 50 µg/mL of imidacloprid with 10 µg/mL of zymosan A (Figure 3C). Ultimately, the effect of imidacloprid and amitraz extended to the rate of NO production. With the observable cut down of NO production after exposure to imidacloprid and amitraz, honeybees would be more susceptible to infection not just at the instant of infection but post-infection as well, especially when considering the downstream signaling of NO and its involvement in the immune response [52,53]. Immune activation by 1 µg/mL zymosan A resulted in a synergistic effect with imidacloprid and amitraz. However, immune activation by 10 µg/mL zymosan A mitigates the cut-down effect on the rate of NO production. Yet, the higher concentration of zymosan A might be equivalent to the immune stimulation of severe nosemosis but without the associated damage, or it could be equivalent to the presence of gut bacteria in honeybees, which plays a role in the resistance to *Nosema* infection [54,55].

Both pesticides showed no effect on NO production in Schneider-2 cells at 15 min exposure times with no change in the production rate of NO despite any concentration of zymosan in all treatments. In fact, after 120 min of exposure, Schneider-2 cells showed an increase in NO production with the 50I-10A combination with 0 or 1 µg/mL zymosan A and a decrease with the 10I-50A combination at 1 µg/mL zymosan A immune activation. At the higher level of zymosan A, all single pesticide exposures and 10I-50A showed an increase in NO production. The production rate of NO is unchanged in all conditions. This indicates that *Drosophila* responds to stressors in a complex manner at given time points and that Schneider-2 cells are overall less affected by pesticides regarding NO alternations with a strong internal regulation mechanism compared to honeybees.

As for the MB-L2 cells, the effect of imidacloprid and amitraz was restricted to combination exposures but never single exposures. It seems that the effect of pesticides on the primary immune response is time-dependent taking into consideration the level of immune activation. With no zymosan A, the decrease in NO appeared after 120 min and in the production rate between the set time points but not at the first time point of measurement. Similarly, exposure to 1 µg/mL zymosan A decreased NO production but only in the 10I-50A combination at 120 min. The production rate was not significantly changed with immune activation. This could posit that *M. brassicae* becomes more resistant to the used pesticides’ effects on oxidative response when immunologically challenged and it takes multiple stressors/stimulants to have an observable effect at the same level of immune activation.

NO functions independently from the ROS system yet it plays a role in limiting the reactivity of hydrogen peroxide and oxygen radicals to specific cellular sites [52]. This is in consensus with our results concerning honeybee hemocytes. However, other studies suggest that H_2_O_2_ can modulate NOS activity via targeting NF-kB gene sites [32], implying a relationship between ROS and RNS production. In our results, the relative hydrogen peroxide levels were the highest when hemocytes were treated with amitraz without immune activation. This may be due to the ability of amitraz and its metabolites to limit ROS elimination [51]. Lower concentrations of amitraz resulted in a higher level of H_2_O_2_ when compared to higher amitraz concentrations. If H_2_O_2_ was observed alone, the expected effect would be that lower amitraz concentrations produce more ROS-mediated cellular damage in honeybees. However, when considering the production of NO at different time points, it is noticeable that a higher amitraz concentration results in significantly lower NO levels. This may infer that the production of NO may be more crucial in determining the susceptibility of hemocytes to ROS-mediated cellular damage than increased levels of H_2_O_2_ in honeybees (*A. mellifera*) when not immunologically challenged. In addition, when amitraz was present with imidacloprid, immune activation with zymosan A was not required to observe a change in extracellular hydrogen peroxide production in honeybee cells but it was required for Schneider-2 cells. The lowest concentrations of H_2_O_2_ were detected with the pesticide combinations in immune-activated honeybee hemocytes and Schneider-2 cells. The MB-L2 cell line was affected only with no immune activation. In contrast to honeybee and Schneider-2, H_2_O_2_ production increased in MB-L2 cells when exposed to 50 µg/mL amitraz and combinations 10I-50A and 50I-10A but no effect was observed with immune activation.

Cell line cultures may have different cell type constitutions than freshly extracted hemocytes; this is important when addressing the results because there is variability in NO production between different cell types within a population, as seen in honeybees where granulocytes showed a higher production of NO than other hemocytes [28].

Nitric oxide production is an ancient trait originating before the divergence of vertebrates and invertebrates [56], thus, it is possible that the utilization of such a molecule is variable and diverse between different species with its multiple roles in immune responses, neural responses, development, and oxidative stress modulation. This variability in responses between different orders may obligate the use of the exact organism in toxicological studies. Honeybee sensitivity to risk factors that contribute to colony collapse disorder is evidently dependent on the combination of specialized characteristics. These characteristics include the genetic diversity of immune genes and detoxifying enzymes, social immunity, the effect of behavioral alterations, and the temperature-dependent effect of pesticides on survivability; these are all considered when studying insects. In this study, we shed light on the highly variable oxidative response of hemocytes between Hymenoptera, Diptera and Lepidoptera candidates.

The oxidative immune response of the European honeybee (*A. mellifera*) is more altered by amitraz and imidacloprid exposure than *D. melanogaster* and *M. brassicae*. Considering that the overall significant effect was apparent with the combinations that included both pesticides, amitraz may indeed prove to decrease the fitness of honeybee colonies in response to diseases [4]. The synergistic effect of amitraz with pesticides was shown before and the order of their exposure was important in elucidating this effect [23]. Thus, the ability of honeybees to resist diseases such as nosemosis and pests such as the wax moth may be hindered by pesticide exposure even if the used concentrations are sub-lethal.

Indeed, our findings suggest a differential effect of risk factor exposure on the immune response in different species representing different orders. In addition, it is recommended to re-evaluate the risks of pesticide exposure to honeybees on all fronts in a comparable manner with the specific conditions. Cox-Foster et al. [12] already presented a meta-genomic survey to assess the degree of contribution of different risk factors to colony collapse disorder. However, temperature changes [57], the order of exposure to risk factors, and its duration are also required to have a comprehensive view. Since CYP450 production affects imidacloprid toxicity [19,20] and metabolism affects amitraz toxicity [51], it is implied that the effect of imidacloprid and amitraz co-exposure may be different from the effect of single exposures, as found in this study. The presented differential effect of pesticides between different insects, and extrapolation of experimentation on “model” organisms may be rendered inadequate depending on the basis of the design and the factors implicated, especially when compared to the honeybee system.

The in vitro system is advantageous where effects and mechanisms concerning pesticide exposure and immune response can be studied in parts that may illustrate the effects observed at the whole organism level. Another advantage is that cell lines are regularly maintained and available when needed but when it comes to honeybees, limitations must be considered. Though freshly collected hemocytes pose a more realistic approach for in vitro assays, extracting hemocytes from honeybee larvae is limited to the seasonal availability of larvae, the physical effort that comes with hemocyte extraction— and that extracting hemolymph from larvae has a risk of contamination with each larva— and it is time-consuming. Not to mention, adult honeybees do not make a preferred alternative to larvae as only a small amount of hemolymph can be extracted from adults compared to larvae [34,58], limiting experimental implementation.

Considering all the previously mentioned points, factors such as co-exposure to different biotic and abiotic factors require extensive studies on honeybees at immunologic, neural, behavioral, and developmental levels. Thus, cell-based-system methodologies must be continued by in vivo application and field applications. The social construct must also be taken to account when deriving the effects of pesticides on the colony level with the inclusion of honeybee drones whose haploid genome may be more sensitive to these risk factors, leading to a weak colony [59].

The obtained results present an observable effect of imidacloprid and amitraz on the immune activation of hemocytes of Hymenoptera, Diptera and Lepidoptera representatives. A complex interaction between these pesticides was also observed with different ratios of exposure. The exposure duration poses a synergistic or antagonistic effect depending on the parameter studied which requires an in-depth understanding of the mode of action in each species. In addition, pesticide exposure indeed altered the immune response and immune activation in insect hemocytes, a point to be taken in earnest when considering the impact of pesticide usage on insects and ultimately biodiversity. In spite of any comparative approaches, the effect of pesticides on honeybee immunity is prominent and requires further evaluation of pesticide usage and its implication in causing CCD in order to preserve the economic and environmental benefits of honeybees.

## 5. Conclusions

Nitric oxide production is not affected by the level of extracellular hydrogen peroxide production in honeybees, fruit fly, or cabbage moth larval hemocytes. The production of nitric oxide and hydrogen peroxide in freshly extracted honeybee hemocytes is more affected by amitraz and imidacloprid exposure than the established cell lines Schneider-2 and MB-L2 of *D. melanogaster* and *M. brassicae,* respectively. The effect of imidacloprid and amitraz on the oxidative response persists after early immune stimulation in hemocytes of the used insect species.

## Figures and Tables

**Figure 1 insects-14-00174-f001:**
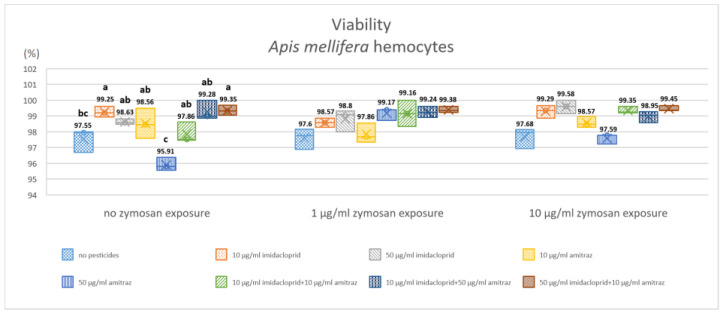
Effect of pesticide exposure on honeybee hemocyte viability after 18 h of exposure. Three groups of immune activation by zymosan A were exposed to different pesticides in single exposures or combinations. Results are expressed in percentages and different letters indicate the significant differences within zymosan groups (*p* < 0.05, *n* = 3).

**Figure 2 insects-14-00174-f002:**
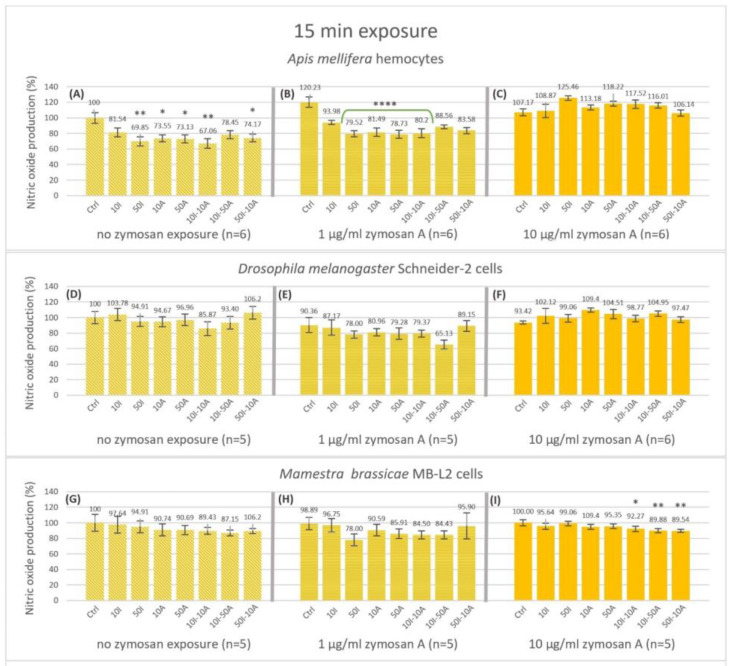
Nitric oxide production after 15 min of exposure of non-activated and activated hemocytes of *Apis mellifera* larvae (**A**–**C**), Schneider-2 cell line (**D**–**F**), and MB-L2 cell line (**G**–**I**), to single or combined exposure to imidacloprid and amitraz. In addition to the no-pesticide controls (Ctrl), hemocytes were exposed to 10 or 50 µg/mL concentrations of imidacloprid (10I, 50I, respectively) or amitraz (10A, 50A, respectively). Three pesticide combinations of 10 µg/mL Imidacloprid + 10 µg/mL amitraz (10I-10A), 10 µg/mL Imidacloprid + 50 µg/mL amitraz (10I-50A), or 50 µg/mL Imidacloprid + 10 µg/mL amitraz (50I-10A). All pesticide exposure conditions included immune activation with two concentrations of zymosan A (1 and 10 µg/mL) or none. Results were expressed as normalized percentages relative to the control group without pesticides or immune activation. Significant differences were tested by Dunnett’s test within zymosan treatment groups relative to the respective controls and designated by an asterisk (* *p* < 0.05, ** *p* < 0.01 and **** *p* < 0.001, *n* = 5, 6). Error bars represent standard errors.

**Figure 3 insects-14-00174-f003:**
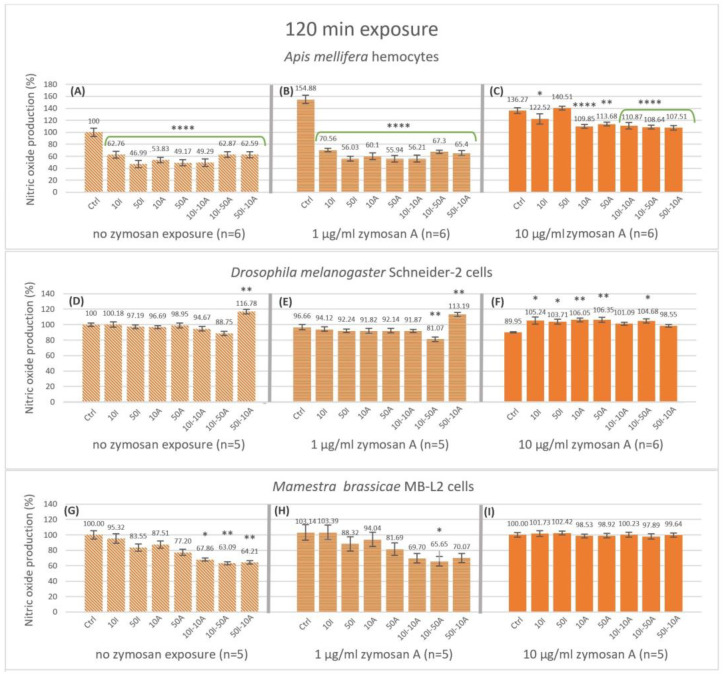
Nitric oxide production after 120 min of exposure of non-activated and activated hemocytes (*Apis mellifera* larval hemocytes, Schneider-2 cell line, and MB-L2 cell line) to single or combined exposure to imidacloprid and amitraz. Hemocytes of *A. mellifera* larvae (**A**–**C**), Schneider-2 cells (**D**–**F**), and MB-L2 cells (**G**–**I**) were exposed to 10 or 50 µg/mL concentrations of imidacloprid (10I, 50I, respectively) or amitraz (10A, 50A, respectively) in single exposures and in combinations of 10 µg/mL Imidacloprid + 10 µg/mL amitraz (10I-10A), 10 µg/mL Imidacloprid + 50 µg/mL amitraz (10I-50A), or 50 µg/mL Imidacloprid + 10 µg/mL amitraz (50I-10A). All pesticide exposure conditions included a control with no pesticide treatment and immune activation with two concentrations of zymosan A (1 and 10 µg/mL) or none. Results were expressed as normalized percentages relative to the control group without pesticide or immune activation. Significant differences within zymosan treatment groups relative to the respective controls were designated by an asterisk (* *p* < 0.05, ** *p* < 0.01, and **** *p* < 0.0001, *n* = 5 or 6). Error bars represent standard errors.

**Figure 4 insects-14-00174-f004:**
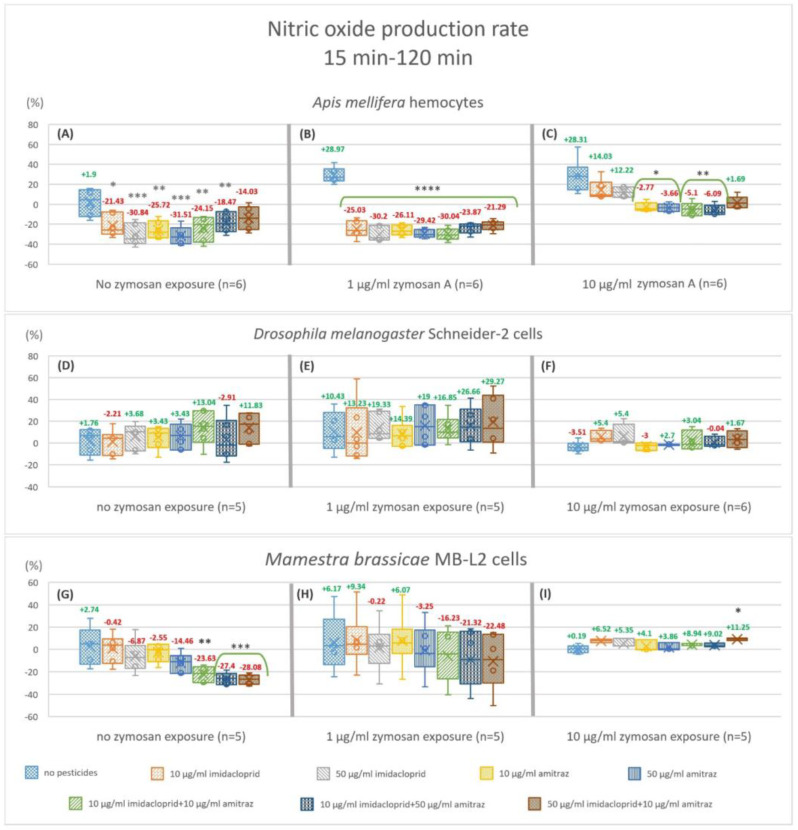
Nitric oxide production fold changes from 15 min to 120 min of exposure. The box plots represent the ratio of relative nitric oxide percentage of endpoint fluorescence measurements at 120 min over 15 min in hemocytes of *Apis mellifera* (**A**−**C**), Schneider-2 cells (**D**−**F**), and MB-L2 cells (**G**−**I**). All results were normalized to the no treatment control (**A**). Statistical analysis was carried out within zymosan A treatment groups. Significant differences within zymosan treatment groups relative to the respective controls are designated by an asterisk (* *p* < 0.05 and ** *p* < 0.01, *** *p* < 0.001 **** *p* < 0.0001, *n* = 5 or 6).

**Figure 5 insects-14-00174-f005:**
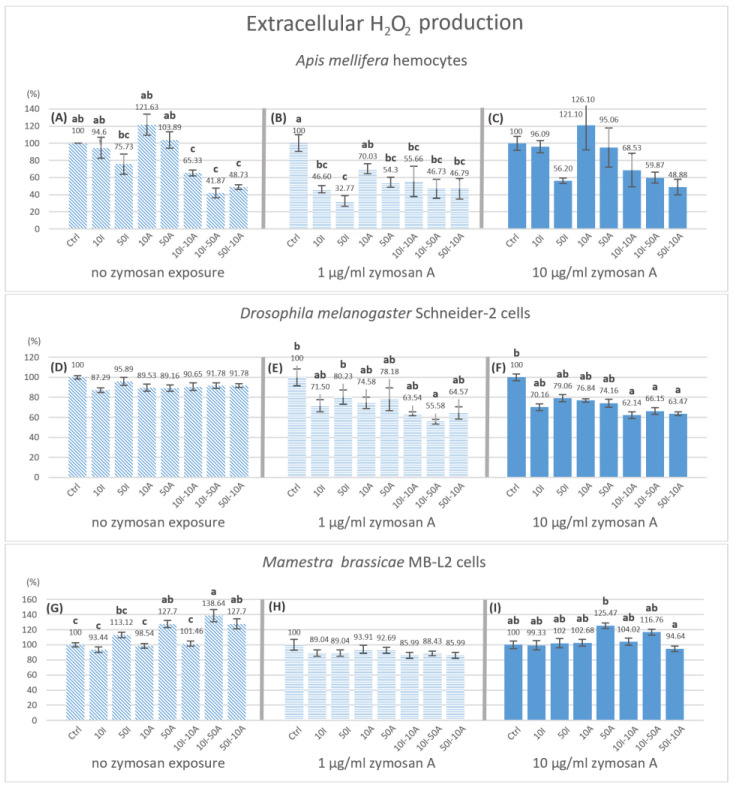
Hydrogen peroxide production by hemocytes after 3 h of pesticide exposure with different levels of immune stimulation. Bar graphs represent the relative hydrogen peroxide concentration produced hemocytes of *Apis mellifera* (**A**−**C**), Schneider-2 cells (**D**−**F**), and MB-L2 cells (**G**−**I**) after 3 h of exposure to pesticides. All concentrations are expressed in µg/mL of imidacloprid (I) or amitraz (A) in legends. Statistical analysis was carried out with in zymosan A treatment groups. ANOVA (Tukey HSD) or Kruskal–Wallis were used to test for significant differences at *p* < 0.05 with *n* = 5. Different letters signify significant differences. The absence of indication refers to no significant differences. Error bars represent standard errors.

## Data Availability

Data produced by this research are available upon request.

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
