# Peer review of "Differential Production of Nitric Oxide and Hydrogen Peroxide among Drosophila melanogaster, Apis mellifera, and Mamestra brassicae Immune-Activated Hemocytes after Exposure to Imidacloprid and Amitraz"

_insects, 2023, doi:10.3390/insects14020174_

Round 1

Reviewer 1 Report

The manuscript attempts to elucidate NO and H2O2 production by hemocytes of D.melanogaster, A. mellifera and M.brassicae upon exposure to imidacloprid and amitraz following immune activation by Zymosan A. I had several issues with understanding the hypothesis that was being tested as well as the method being employed. The Introduction starts off with Colony Collapse Disorder in Honey bees and then the narrative changes to a comparative study of nitric oxide and hydrogen peroxide production in hemocytes in a Dipteran (D. melanogaster), a lepidopteran (M. brassicae) and a hymenopteran (A. mellifera). This is where the entire premise of the study gets confusing, what exactly do the authors want to investigate? You cannot compare apples and oranges!

I was pretty much disappointed with this manuscript for the following reasons:

1. The idea starts off with CCD and then veers off into something that is quite different. Have the authors checked in hives which have been infected with a pathogen and examined the NO and H2O2 production in the infected individuals? That would at least have served as a starting point to this investigation.

2. What was the idea of using Drosophila cell lines and M. brassicae cell lines to this study? The immune response elaborated by different insect species will be different - what else is expected? You cannot compare NO and H2O2 production levels and rates between two different species unless you make sure that the cells you are examining are similar. The authors themselves state that there is a difference in the hemocytes between these two species, if so, their response to a similar stimulus will also vary! Importantly, the authors make no attempt to normalize the cell numbers (i.e. number of hemocytes that were being exposed to Zymosan-A, or to the pesticides). This shows that the authors have a very inappropriate experimental strategy!

3. It would have made more sense to just stick to the NO and hydrogen peroxide production of hemocytes of A. mellifera and its differential response to two different pesticides. The work could have been further expanded upon by making parallel field-level studies to mimic the in -vitro studies being conducted.

4. To me this looks like the authors have generated data, but then try to put it all together to make sense, which does not seem a sensible way to conduct research! What is it that the authors want to investigate?

5. What is the read-out for immune activation by Zymosan-A?

6. If this work is to get published, I highly recommend omitting the data related to D. melanogaster and M. brassicae. These have no relevance for the present study and its overall objective.

7. Two of the cell lines used are established cell lines, to compare it to hemocytes from A. mellifera which has been freshly extracted does not make sense!

8. I would be interested to see the molecular mechanisms behind the activation of NO, what is the NOS expression level in A. mellifera? Does the NO being produced actually have an antimicrobial effect or does it have a cytotoxic effect?

9. The present version of the manuscript is poorly thought-out and crafted. Its important to have a clear-cut hypothesis and an experimental plan that is logical. Just generating random data does not guarantee its publication as a scientific manuscript!

Author Response

  1. It is not feasible to measure NO or H2O2 at the beginning of the immune response at the individual level with the 24 used conditions for the following reasons:

1.1. Each bee would have to be exposed to the treatment for exactly the same amount of time, which is not plausible, and the hemocytes taken directly from each individual bee/larvae. The process of extracting hemocytes is time-consuming and is not convenient to determine NO or H2O2at the needed time points.

1.2.  With the in-vitro approach, the hemocytes are pooled and normalized in concentration allowing a comparison of results between different treatments. Hemocyte concentration cannot be normalized if taken from individuals for direct measurement and it will affect the fluorimetric and spectrophotometric measurement.

2. Since all used species are insects are are implicated in the pollination process to different extents and use the same pathways of the immune system to defend against pathogens it is plausible to compare them to see the effect of pesticides. The importance of using cells from different species is to know if the same pesticides have different incidences on different species and to see if honeybees are more or less susceptible to the use of pesticides compared to other species in the same environment. We add that the use of cell lines in continued in toxicological studies as model systems and the generated data is to provide more information not just to comparative studies but also to studies concering the use of these cell lines themselves.

3. As stated in (1), it is not feasible to conduct field studies with the used parameter and the designated time points of measurement

4. The cells of the species used represent insect orders that cohabit the same environment and are exposed to the same risk factors. We aim to asses the effect of pesticides on the immune reponse of each organism to evaluate their susceptibility to diseases and to compare the difference of susceptibility to asses an advantage of one over the other especially in the case of honeybees which have a high economic and environmental value. not to mention that some pests of honeybees belong to the Lepidoptran order, as advantage of these pests in the same environment may well put honeybees at higher risk of colony death.

5. Zymosan A activation without pesticides is a reference point to be normalized and compared to other treatments within the same group having the same zymosan concentration.

6. As we are taking a comparative approach among insect orders as part of the theme to this study, we would find it more informative to keep the data from the used cell lines.

7. The difference between cell lines and freshly extracted cells is indeed a possibility for the variation; however, cell lines themselves are used as a reference to the original organism despite any variation caused by their establishment. Thus, their inclusion in this study and their comparison to freshly extracted hemocytes comes from a similar concept.

8. We aim to assess the cellular response with unified in-vitro assay approach as the first level of analysis. The gene expression analysis will be reserved for the next level of similar studies after more data is generated at the cellular level in order to have a more in-depth understanding of their mechanism in further advancements

9. The article has been optimized according to the reviewers’ comments for better presentation

Reviewer 2 Report

The article has a scientific value and novelty. I have read the article with interest and pleasure. The introduction describes an actual state of the art in this field, some statements should be covered with proper publications. Methodology is good but some explanation should be provided: Hemocyte extraction, lines 111-113 authors should separate hemocyte extraction, viability tests, hydrogen peroxidase analysis and nitric oxide analysis.

Introduction: there should be added data about zymosan and its impact on insects; there should be emphasis that now we face the global extinction problems and the presented research contributes to saving insects esppecially exposed to pesticide (please add info about mass extinction from: “A Short Guide to the Sixth Mass Extinction – is the Anthropocene an extended suicide?” Revista de Educacion, 395 (1): 27-41).

The Discussion should be extended with one paragraph of general conclusions.

Author Response

The methodology has been modified as required and a section regarding mass extinction has been integrated into the introduction from the reference well proposed by the reviewer. In addition, we added more information on zymosan's effect on the immune response in insects

The discussion was extended as requested

Reviewer 3 Report

(1)    According to the instructions for Authors, the abstract should be a total of about 200 words maximum, so please shorten the summary to about 200 words.

(2)    Line 137: Section 2.4: the powdery compound of imidacloprid can be directly dissolved in Dulbecco’s Phosphate Buffered Saline (PBS; D8537, Sigma-Aldrich™) ?

Author Response

(1) The abstract is reduced as requested

(2) Imidacloprid and zymosan were sonicated in PBS/culture medium to allow them to dissolve. We added the sonication to the methodology

Reviewer 4 Report

The manuscript "Differential production of nitric oxide and hydrogen peroxide among Drosophila melanogaster, Apis mellifera, and Mamestra brassicae immune-activated hemocytes after exposure to imidacloprid and amitraz is a well-designed study. I have some minor comments/reviews.

1. Line 31-34. Page 1. More clarity is needed. This paragraph is confusing. Some corrections are marked in red.

We measured the effect of these exposures on cell viability and nitric oxide (NO) production from 15 until 120 minutes to evaluate the toll of pesticides on primary immune response and on extracellular hydrogen peroxide (H2O2) production after 3 hours to assess potential diseases susceptibility by the low oxidative response and probable oxidative damage.

2. Line 32, page 2. Other than neonicotinoids, amitraz is an acaricide insecticide.

Add and between acaricide insecticide.

 3. Line 46, Page 2 ….in insects like the mosquito

Change to in insects like mosquitoes.

 4. Line 57-68 Review syntax and time

 5. Consider reducing the introduction; It is too long.

 6. Line 131. Did you use active ingredients of both insecticides? Please, specify.

 7. Throughout the manuscript, the name of the species should be in italics

 8. Lines 258-262 The paragraph is confusing

Author Response

1-4. All requested modification are made accordingly 

5. The introduction has been modified; the paragraph describing the types of hemocyes has been removed and some information on species extinction and the effect of zymosan have been added as requested by the other reviewers. 

6. We did use the active ingredients and it is now stated in the materials and methods as requested

7. All insect latin names are now converted to italics 

8. The paragraph is now re-explained clearly

Round 2

Reviewer 1 Report

I am still not satisfied by the responses the authors have provided to my comments/ suggestions.

1. The basic premise of this work appears to be rooted to the CCD in honey bees and so when the authors start off by introducing this concept, why would it not be feasible to check for NO and hydrogen peroxide production in bee samples from hives infected/ affected? That would lend more credence to the subsequent in vitro investigations! I am not asking to expose the bees to same amount of pesticides, just to strengthen your original hypothesis that NO and hydrogen peroxide do get produced as a defensive response in field level situations.

2. In response to my comment the authors state that all species used in this study have an impact on pollination. Can you cite concrete data to support this statement? Also, why would you think that the response to pesticides would be same among different insect species? This is where I have a problem understanding the line of thinking of the authors. Each of these insects is different from the other, if you had been comparing within an order, I would understand, here that is not the case!

3. I am not satisfied with this explanation. Why is it not feasible? The ultimate purpose of this entire investigation is to throw light on the field level situations, if such a study is not feasible at the field level, then what is the purpose of the in vitro study?

4. The authors do not clarify what is the read-out for immune priming by Zymosan A - I mean what test do they do to show that it is indeed activated? Just the NO and hydrogen peroxide production?

5. I am not satisfied with the authors explanation of comparing freshly prepared hemocytes with established cell lines. There are bound to be differences which seem to be conveniently ignored by the authors.

6. I still stand by my review and recommend to reject this version of the manuscript unless this manuscript is bolstered by field level data, and the focus is on only honey bees, the current manuscript as it stands is very poorly designed and executed.

Author Response

1. There are advantages and disadvantages when using the in-vitro system or field study, each with its limitations that may limit similar applicability with the same approach in either.

It is not feasible to measure NO or H2O2 from hives with the used time points or methodology due to the stated obstacles in the application. The in-vitro system posed a by-pass to that limitation of the field study. Hence, a field experiment is not plausible with the exact parameters used in the article.

The presented results could be the basis for subsequent studies with different approaches or parameters for cell-based assays or field studies with no necessity for the exact type of measurement.

To add, since we took a comparative approach between species we needed to compare on the same level (using cells was more convenient for this approach due to their immediate availability and easier manipulation).

Previous studies have been made to indicate the effect of pesticides on the immune/oxidative response in field conditions or in-vivo. but with a different approach and different parameters.

https://link.springer.com/article/10.1007/s13592-018-0583-1#Sec15 https://link.springer.com/article/10.1007/s13592-014-0308-z

https://link.springer.com/article/10.1007/s13592-018-0583-1

https://www.ncbi.nlm.nih.gov/pmc/articles/PMC7290330/

https://www.pnas.org/doi/10.1073/pnas.2011828117

https://pubmed.ncbi.nlm.nih.gov/35663981/

2. The following articles contain information on the implication of the used species/orders in pollination

https://doi.org/10.1093/ee/nvab145  (M. brassicae)

https://doi.org/10.1111/j.1365-2311.2010.01247.x (Lepidoptera)

https://www.sciencedirect.com/science/article/pii/S0960982210011516 (D. melanogaster)

https://www.ncbi.nlm.nih.gov/pmc/articles/PMC4549958/ (Drosophila).

In addition, we restate that we are using drosophila hemocytes as a comparative model and MB-L2 as representatives of the Lepidoptera in order to assess any advantage of the honeybee pest, the Lepidopteran wax moth.

The authors aim to assess the difference in response between the species. The mentioned similarity was referring to the immune pathways among these insects and not the similarity in the resulting responses.

3. Further investigations on the field level do not require a similar approach. Thus, the limitation of the experiment to the cell-based system does not affect other experiments to get a more comprehensive view. However, as stated, this approach is limited to the in-vitro system with these conditions. The purpose of this study is to give insight into the specific cellular level which is unobservable in an in-vivo system where more complex interactions occur. A specific observation as a part of a whole and complementary to it despise any limitations.

4. We believe that NO and H2O2 were sufficient indicators of immune activation by zymosan A. Evidence supporting immune activation by zymosan in insects was already presented in previous studies; thus, its effect was a given in regard to the experiment.

https://www.sciencedirect.com/science/article/abs/pii/0022201168900748

https://link.springer.com/chapter/10.1007/978-3-642-76074-7_24

More indications can be found in the book:

insect infection and immunity; Evolution, Ecology, and mechanisms by Rolf and Reynolds

5. In the same way that cell lines are used as representatives of the organism. We believe that comparing cell lines and freshly extracted cells is as valid (via substitutive comparison). The validity of using cell lines as representatives comes from their continuous use in research.

6. With the latter, we hope the reviewer would reconsider their opinion on the article

Round 3

Reviewer 1 Report

Unfortunately we seem to be at an impasse. The authors are resistant to revise the manuscript or provide a cogent response to my questions.

I am still not satisfied with the authors statement that Nitric oxide production cannot be measured in field level conditions in honey bee colonies which have Varroa infection or have been exposed to pesticides. If this is the case, then the work is conceptually flawed since the central idea is to show how various insects/ pollinators respond via their immune response to pesticides.

The experimental design has to support field level scenario based on results obtained from in-vitro lab based experiments. The authors categorically state that this is not possible. If so, what is the purpose of such a study?

I would like to formally withdraw from the review process since it appears the authors are not comfortable with revising the manuscript. In turn, I cannot in good faith recommend publication of a manuscript that I believe to be conceptually incorrect.

Author Response

As stated, this study can be continued to address CCD with complimentary approaches on the field level and not necessarily the same parameters as requested by the reviewer. 

We thank the reviewer for the time and effort taken on the review process.

All suggestions given will be taken to consideration in the continuation of this work. However, for this study we find it is best to abide by modifications that may not cause a clash with the previous reviewers' opinions and suggestions.